# Bronchoscopy Findings during Percutaneous Dilation Tracheostomy: A Single Tertiary Medical Center Experience

**DOI:** 10.3390/diagnostics13101764

**Published:** 2023-05-17

**Authors:** Ko-Wei Chang, Hsin-Yueh Fang

**Affiliations:** 1Department of Thoracic Medicine, Chang Gung Memorial Hospital, Taoyuan 333, Taiwan; b9302072@cgmh.org.tw; 2Graduate Institute of Clinical Medical Sciences, College of Medicine, Chang Gung University, Taoyuan 333, Taiwan; 3Division of Thoracic Surgery, Chang Gung Memorial Hospital, Taoyuan 333, Taiwan

**Keywords:** bronchoscopy, intra-airway abnormality, tracheostomy

## Abstract

Percutaneous dilation tracheostomy (PDT) is a common procedure in intensive care units. Bronchoscopy has been recommended to guide PDT to decrease complication rates, but no study has analyzed bronchoscopy outcomes during PDT. In this retrospective study, we analyzed bronchoscopy findings and clinical outcomes during PDT. We collected data on all patients who underwent PDT between May 2018 and February 2021. All PDT operations were guided by bronchoscopy, and we assessed the airway to the third order of the bronchi. Forty-one patients who underwent PDT were included in this study. The average duration of PDT was 102.8 ± 34.6 s, and the average duration of bronchoscopy was 49.8 ± 43.8 s. No complications related to bronchoscopy and no significant changes in gas exchange or ventilator parameters were noted after the procedure. Fifteen patients (36.6%) exhibited abnormal bronchoscopy findings, including two patients (13.3%) with intra-airway mass lesions and obvious airway obstruction. None of the patients with intra-airway masses could be liberated from mechanical ventilation. This study observed a non-negligibly high incidence of unexpected endotracheal or endobronchial masses in patients with chronic respiratory failure during PDT, and a high rate of weaning failure was noted in these patients. The completion of bronchoscopy during PDT may provide additional clinical benefits.

## 1. Introduction

Tracheostomy is now a common procedure, particularly for patients who require prolonged usage of mechanical ventilation [1,2]. Tracheostomy is indicated for long-term ventilator use, weaning failure, upper airway disease, and copious secretions [2]. In addition to surgical tracheostomy, percutaneous dilatational tracheostomy (PDT) is commonly used in the intensive care unit (ICU) [3]. Numerous studies have compared surgical tracheostomy with PDT. A meta-analysis demonstrated that PDT can reduce wound infection, bleeding, and mortality [4], and another meta-analysis revealed that PDT can be performed faster but with greater technical difficulties [5]. Guidelines from 2018 describe PDT as the standard method in the ICU (grade 1+/strong agreement) [1]. The use of bronchoscopy to guide PDT has been suggested to potentially reduce complication rates [6].

Chronic respiratory failure can be caused by pulmonary, neurological, or cardiovascular diseases. Among these, pulmonary disease is the most common cause [7,8]. No study on patients with chronic respiratory failure has discussed bronchoscopy findings. Although studies have suggested tracheostomy for prolonged mechanical ventilation [1], the weaning success rate is not significantly higher in patients undergoing tracheostomy [8].

This study analyzed the data of patients with chronic respiratory failure who underwent PDT guided by bronchoscopy to investigate the relationship between bronchoscopy and weaning outcomes.

## 2. Materials and Methods

### 2.1. Patient Population

We conducted a retrospective case series study in a tertiary referral center in Taiwan with 278 adult ICU beds and 24 respiratory care center beds. The respiratory care centers specialize in the care of patients who require prolonged ventilation and have no other systemic problems. We collected the data of all patients who underwent PDT between May 2018 and February 2021. The demographic data, bronchoscopy findings, laboratory data, pathology reports, ventilator settings, and clinical conditions and outcomes were all collected from the electronic medical record system. The sequential organ failure assessment (SOFA) score [9], laboratory data, ventilator parameters, and blood gas data were collected within 3 days before PDT. Institutional review board approval was obtained (CGMHIRB-202101105B0).

### 2.2. PDT

Our PDT team included a thoracic surgeon and a pulmonologist. In our hospital, the bronchoscopy is mandatory to guide the PDT to avoid complications, such as misplaced puncture, posterior tracheal wall damage, or hemorrhage, and we arranged bronchoscopy examination for all patients who underwent PDT. Patients who were indicated for tracheostomy exhibited long-term ventilator usage, weaning failure, upper airway disease, or copious secretions. Contraindications for PDT include a high bleeding risk that may necessitate electrocoagulation, abnormal neck structure, or previous neck surgery; some patients did not undergo PDT due to the duty doctor’s preferences. Before the surgical procedure, the pulmonologist assessed the airway to the third order of the bronchi using bronchoscopy, and secretions in the trachea and bronchi were cleaned thoroughly. Subsequently, the endotracheal tube was drawn near the vocal cord level, and a bronchoscope was placed in the upper trachea to monitor the puncture site. The thoracic surgeon then performed the PDT as the standard procedure [3]. We used the Ciaglia Blue Rhino G2 set (Cook Medical, Bloomington, IN, USA) for all of the procedures in this study. The bronchoscope was inserted into the tracheostomy tube to assess bleeding and remove as much blood or secretions as possible. Bronchial washing for pathogen survey or biopsy by forceps was performed after the tracheostomy under the bronchoscopy findings and the request for duty intensivists. All of the PDTs or bronchoscopies were performed by the same thoracic surgeon and pulmonologist, so factors such as the operator’s skill or practice habits may not have impacted further analysis in this study. Bronchoscopy duration was measured from the entrance of the endotracheal tube to the completion of the bronchoscopy survey. Tracheostomy duration was measured from needle puncture to the completion of the procedure.

### 2.3. Arterial Blood Gas and Ventilator Setting Data

The respiratory therapists and intensivists established and adjusted the mechanical ventilator settings according to the patient’s condition and arterial blood gas data. All patients were administered mechanical ventilation in pressure-limited ventilation, and the time-cycled pressure-control mode or flow-cycled pressure-support mode was chosen according to the patients’ demand for a mechanical ventilator. The peak airway pressure and positive end-expiratory pressure were set, and the tidal volume was detected by the ventilator. If the patients used non-invasive ventilation or a T-piece before the tracheostomy, the intubation was performed before the operation. Arterial blood gas and ventilator setting data were collected in the morning preceding PDT and the morning following PDT. 

The dynamic lung compliance was calculated as tidal volume divided by dynamic driving pressure, and the dynamic driving pressure was calculated as peak airway pressure minus positive end-expiratory pressure. The changes in arterial blood gas data or ventilator setting data were defined as post-tracheostomy data minus pre-tracheostomy data.

### 2.4. Statistical Analysis

The numbers (percentages) for nominal variables are presented, and Pearson’s chi-square or Fisher’s exact tests were used to compare these variables. The means ± standard deviations of continuous variables are presented, and Mann–Whitney U tests were used to compare these variables. Paired samples t-tests were used to compare the parameters before and after tracheostomy. Two-tailed tests were used, and the significance level was set at *p* < 0.05. All analyses were conducted using SPSS version 22.0.0 (IBM, Armonk, NY, USA).

## 3. Results

### 3.1. Demographic Data

Forty-one patients underwent PDT performed by our team during the specified period, all of whom were included in this study. A flow chart of the patients in this study is shown in Figure 1. The demographic data are presented in Table 1. Twenty-two patients were female. The mean age of the patients was 70.3 years. Most patients underwent tracheostomy in the medical ICU or respiratory care center (73.2%), and the remaining patients were admitted to the cardiac, alimentary, or surgical ICUs. Twenty-seven patients (65.9%) had lung disease-induced chronic respiratory failure, six patients had central nervous system disease-related respiratory drive problems, and eight patients had central airway problems (i.e., vocal cord palsy in two patients, supraglottic inflammation in one patient, facial bone fracture in one patient, and trachea problems in four patients). The average duration from the start of mechanical ventilation to the tracheostomy was 26.4 days. Before undergoing tracheostomy, 23 (56.1%) patients used a ventilator in pressure control mode, thirteen (31.7%) patients used a ventilator in pressure support mode, two patients used noninvasive positive pressure ventilation, one patient used a T-piece, and one patient used a simple mask. The mean PaO_2_/FiO_2_ ratio was 313.5 mm Hg, and the mean PaCO_2_ was 45.3 mm Hg. The mean SOFA score before tracheostomy was 4.6.

### 3.2. Clinical Outcomes

The average duration of PDT was 102.8 ± 34.6 s (minimum–maximum duration was 61–229 s), and no significant differences between patients with intra-airway abnormalities and other patients were observed (98.9 ± 39.0 vs. 105.5 ± 31.9, *p* = 0.334). The average duration of bronchoscopy was 49.8 ± 43.8 s (minimum–maximum duration was 13–242 s); bronchoscopy duration was significantly longer in patients with intra-airway abnormalities than in other patients (77.3 ± 57.3 vs. 33.1 ± 20.6 s, *p* = 0.001; Table 1). No complications related to bronchoscopy, such as significant oxygen desaturation pneumothorax, or bleeding, were recorded during the examination. Tracheostomy wound blood oozing was observed in two patients, but this was resolved after local compression and epinephrine gauze administration. None of the procedures involved any injury of the thyroid ima artery, a variant artery supplying the thyroid that exists in 3.8% of the population [10].

Of these patients, 19 patients (46.3%) were liberated from mechanical ventilation, 14 (34.1%) remained ventilator-dependent, and 8 (19.5%) died in the hospital. 

### 3.3. Comparison of Gas Exchange and Ventilator Parameters before and after Tracheostomy

The gas exchange and ventilator parameters data before and after tracheostomy are shown in Table 2. The PaO_2_/FiO_2_ ratio increased after PDT, but there was no significant difference (from 313.5 ± 100.4 to 331.6 ± 93.8 mm Hg, *p* = 0.196). The PaCO_2_ measure (from 45.3 ± 13.1 to 45.4 ± 12.0 mm Hg) and pH level (from 7.4 ± 0.1 to 7.4 ± 0.1) exhibited no significant differences before and after tracheostomy. In patients with mechanical ventilation before tracheostomy, no significant changes were noted in peak airway pressure (from 23.4 ± 6.9 to 22.6 ± 7.8 cm H_2_O, *p* = 0.403), positive end-expiratory pressure (from 8.1 ± 0.8 to 7.9 ± 1.0 cm H_2_O, *p* = 0.183), tidal volume/predicted body weight (from 8.3 ± 1.9 to 8.3 ± 2.7 mL/kg, *p* = 0.962), or dynamic lung compliance (from 36.5 ± 23.4 to 39.0 ± 24.9 mL/cm H_2_O, *p* = 0.455).

### 3.4. Bronchoscopy Findings

Fifteen patients (36.6%) exhibited abnormal bronchoscopy findings, including seven patients (46.7%) with increased secretion in the distal airway (one patient contracted Acinetobacter baumannii from the bronchial washing culture) (Figure 2a), three patients (20.0%) with endobronchial granulation tissue without airway obstruction (Figure 2b), one patient (6.7%) with mucosal patches that a secretion culture indicated to be Burkholderia cepacia complex (Figure 2c), one patient (6.7%) with peripheral airway mucosal blood oozing (Figure 2d), one patient (6.7%) with known tumor invasion, and two patients (13.3%) with intra-airway mass lesions with obvious airway obstruction from suspected malignancies. The brief case histories of these two patients were as follows:

#### 3.4.1. Case 1

A 92-year-old man was admitted with dyspnea, pneumonia, and respiratory failure. Tracheostomy was performed on ventilator day 25 for long-term ventilator usage. A bronchoscopy revealed an irregular surface mass lesion in the right middle bronchus (Figure 2e), and a biopsy indicated the presence of atypical cells. However, the patient’s family refused further testing or management, and the patient was transferred to the respiratory care ward.

#### 3.4.2. Case 2

A 60-year-old man with underlying stage-IV squamous cell carcinoma of the lung experienced respiratory failure caused by nosocomial pneumonia. A bronchoscopy on ventilator day 3 indicated only increased secretion and no mass lesions. A tracheostomy was performed on ventilator day 22 because of the difficulty of weaning with high airway resistance. The bronchoscopy revealed an irregular whitish mass in the lower trachea with approximately 50% obstruction (Figure 2f). A thoracic surgeon was consulted for tumor removal or stent placement, but the patient’s family refused further management. The patient was transferred to the respiratory care ward. 

### 3.5. Bronchoscopy Findings and Clinical Outcomes

The clinical outcomes of different bronchoscopy findings are shown in Table 3. The patients with no abnormal bronchoscopy findings had a 50% success rate of weaning from a mechanical ventilator. The patients with increased secretion in the distal airway and endobronchial granulation tissue without airway obstruction exhibited better clinical outcomes than others, and their rates of ventilator liberation were 57.1% and 66.7%, respectively. None of the patients with mucosal patches, intra-airway masses, tumor invasion, or endobronchial blood oozing were liberated from mechanical ventilation, and 80% of such patients required long-term mechanical ventilation.

Patients with no abnormal bronchoscopy findings and with increased secretion in the distal airway, compared to patients with unsolvable abnormalities (endobronchial granulation tissue, tumor invasion, intra-airway mass lesions, or endobronchial blood oozing), had no significant change in the PaO_2_/FiO_2_ ratio after tracheostomy (14.5 ± 87.1 vs. 35.0 ± 94.9 mm Hg, *p* = 0.508). However, patients with no abnormal bronchoscopy findings and with increased secretion had better dynamic lung compliance, and patients with unsolvable abnormalities had poor dynamic lung compliance after tracheostomy (+4.5 ± 21.0 vs. −5.7 ± 7.8 mm Hg, *p* = 0.059).

## 4. Discussion

Several studies have discussed the relationship between PDT and bronchoscopy, but no study has investigated bronchoscopy findings during PDT. In this study, an average of 49.8 s was required for a bronchoscopy survey of the third order of the bronchi, and no obvious complications were noted. We observed intra-airway abnormalities in 36.6% of the patients, and 13.3% of them had unexpected mass lesions that were highly suspected of malignancy in the airway. Patients with intra-airway masses, mucosal patches, or endobronchial blood oozing had a low probability of being liberated from ventilation.

Whether bronchoscopy guidance is required during PDT remains controversial. Studies have suggested using bronchoscopy-guided PDT for safety purposes [11], but some studies have reported no differences in complication rates with or without bronchoscopy [12,13]. Furthermore, one study reported that compared with ultrasound-guided procedures, bronchoscopy may confer a higher risk of hypoventilation, hypercapnia, and respiratory acidosis [14]. Another study reported that the puncture site varied even in the bronchoscopy-guided procedures [15]. However, a randomized-controlled study [16] and a meta-analysis [17] have reported that the complication rates and clinical outcomes of ultrasound-guided PDT were not significantly different from those of bronchoscopy-guided PDT. In our study, the use of bronchoscopy to guide the PDT did not induce a significant change in PaCO_2_ levels compared to the levels before the procedure (from 45.3 ± 13.1 to 45.4 ± 12.0 mm Hg), and no obvious complications were recorded. Moreover, bronchoscopy has additional benefits, such as the assessment of airway conditions, the identification of previously undetected endotracheal or endobronchial abnormalities, and secretion suction. The clinical benefits of secretion suction are difficult to determine because we cannot compare the clinical outcomes if we do not remove the secretion. The collection of unhandled secretion may interfere with PDT monitoring, and we have to clean the secretion as much as possible. However, detected endotracheal or endobronchial abnormalities can reveal the possible etiology of respiratory failure, such as that observed in Case 2, or reveal a possible malignant disease, such as in Case 1.

A study analyzed gas exchange before and after PDT and reported a significant decrease in PaCO_2_ after PDT (from 43 ± 9 to 42 ± 7 mm Hg) without significant changes in the PaO_2_/FiO_2_ ratio in all patients (from 304 ± 85 to 305 ± 81 mm Hg) [18]. Another study reported a considerable increase in PaCO_2_ (24 ± 3 mm Hg) during PDT [14]. In our study, PaCO_2_ exhibited no significant changes (from 45.3 ± 13.1 to 45.4 ± 12.0 mm Hg), and the PaO_2_/FiO_2_ ratio improved after PDT without any statistical significance (from 313.5 ± 100.4 to 331.6 ± 93.8 mm Hg, *p* = 0.196). One explanation for this difference may be the different timing of arterial blood gas sampling or differences in the patients’ baseline conditions. The PDT duration (13 ± 6 min) in one study [14] was much higher than that of our study (sum of bronchoscopy duration and PDT duration, 159.2 ± 65.8 s), and the different procedure duration may influence changes in PaCO_2_. Hence, the changes in PaCO_2_ reported in research are inconsistent, and increases in PaCO_2_ may not be a serious concern in bronchoscopy-guided PDT.

A study investigating pediatric tracheostomy patients evaluated routine airway outcomes approximately one year after tracheostomy [19], and they observed that suprastomal granulation was the most common abnormality. However, they also reported that distal airway granulation was observed in 4.9% of the patients. They noted that the decision of whether to resect granulation was based on the probability that the larger granulation may have obstructed the airway. Of these patients, 61.5% underwent granulation resection. In our study, distal airway granulation was observed in 7.3% of the patients. One of them was removed using forceps after PDT, and the others were simply observed. Two of these patients were later liberated from mechanical ventilation without obvious airway obstruction.

Several etiologies may cause endotracheal or endobronchial masses with airway obstruction, including both malignant and nonmalignant diseases, such as granulation tissues, webs, or pseudotumors [20]. Several case reports have reported endobronchial or endotracheal lesions that induce acute respiratory failure, including lymphoma [21,22], metastatic hepatocellular carcinoma [23], or small-cell lung cancer [24]. However, no study has explored the relationship between endotracheal or endobronchial masses with chronic respiratory failure or ventilator dependence. As routine screening using bronchoscopy for airway mass lesions in patients with chronic respiratory failure is infeasible, we used bronchoscopy-guided PDT to survey the accessible airway.

Another benefit of bronchoscopy during PDT is complete sputum suction. A study used daily bronchoscopic sputum suction in patients with acute exacerbation of chronic obstructive pulmonary disease and respiratory failure compared to general sputum suction, and the results showed the patients with daily bronchoscopic sputum suction had better infectious control, shorter mechanical ventilation duration and hospital stay, lower reintubation rate, lower ventilator-associated pneumonia rate, higher weaning success rate, and lower mortality [25]. Another study on patients with bronchiectasis showed that patients who received bronchoscopic airway clearance therapy had a significantly longer duration to another acute exacerbation after discharge and had fewer symptoms [26]. In our study, we only performed the bronchoscopic sputum suction once during the PDT. However, patients with no endobronchial abnormalities or increased secretion had better dynamic lung compliance after the procedure. Further prospective studies are needed to confirm the benefit of more frequent bronchoscopic sputum suction in chronic respiratory failure.

This study had several limitations. First, only one group at our hospital was qualified to undergo PDT; therefore, the sample size was limited. However, we observed two patients with endobronchial or endotracheal masses even in such a small sample; thus, the number of abnormalities observed is likely to be higher in larger samples. Second, the diversity of patients in our study may have interfered with the analysis of endobronchial abnormality etiologies or predictors of weaning success. Third, the patients in our study were not representative of all the patients with chronic respiratory failure because we only included patients who underwent PDT, and randomization could not be performed. A large-scale cross-sectional study is required to investigate the prevalence of endobronchial or endotracheal abnormalities in patients with chronic respiratory problems, and a randomized controlled design should be used to investigate the relationship between these abnormalities and clinical outcomes.

## 5. Conclusions

We observed a non-negligibly high incidence (4.9%) of unexpected intra-airway masses in patients with chronic respiratory failure during PDT, and they exhibited a high weaning failure rate (100%). The completion of bronchoscopy during PDT did not significantly increase the procedure duration or complication incidence but may provide additional clinical benefits.

## Figures and Tables

**Figure 1 diagnostics-13-01764-f001:**
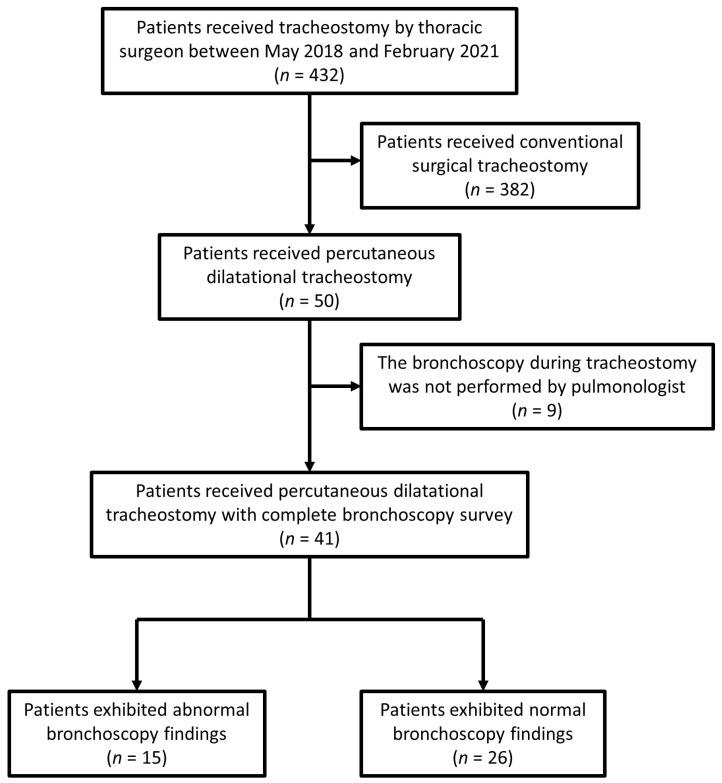
Flow chart of patients in this study.

**Figure 2 diagnostics-13-01764-f002:**
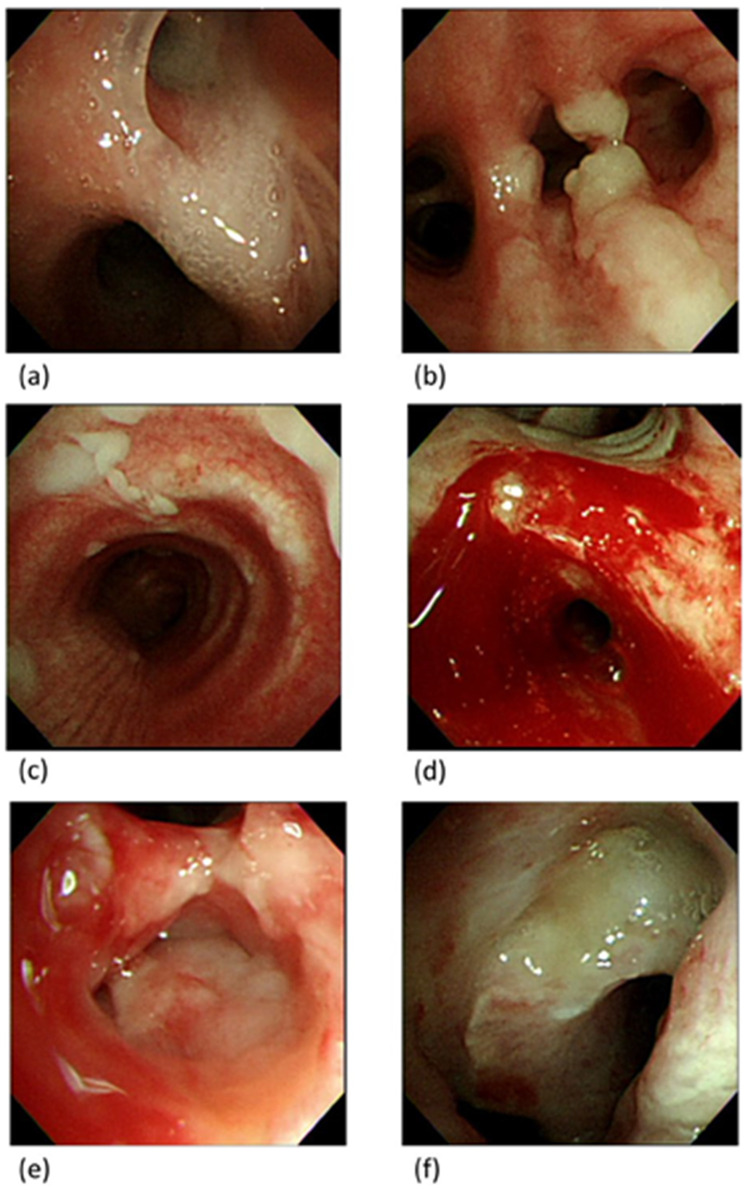
Bronchoscopy findings. (**a**) Increased secretion in the distal airway; (**b**) endobronchial granulation tissue without airway obstruction; (**c**) mucosal patches; (**d**) peripheral airway mucosal blood oozing; (**e**) endobronchial mass; (**f**) endotracheal mass.

**Table 1 diagnostics-13-01764-t001:** Demographic data.

Characteristics	Total Patients (*n* = 41)	Intraairway Abnormality (*n* = 15)	No Intraairway Abnormality (*n* = 26)	*p* Value
Sex (female/male)	22/19	7/8	15/11	0.495
Age (years)	70.3 ± 13.1	69.8 ± 10.7	70.5 ± 14.5	0.758
BMI (kg/m^2^)	22.7 ± 4.1	23.0 ± 4.0	22.5 ± 4.2	0.799
Patients’ source				
Respiratory care center	16 (39.0%)	7 (46.7%)	9 (34.6%)	0.667
Medical ICU	14 (34.1%)	4 (26.7%)	10 (38.5%)	
Cardiac ICU	2 (4.9%)	0 (0.0%)	2 (7.7%)	
Alimentary ICU	6 (14.6%)	3 (20.0%)	3 (11.5%)	
Surgical ICU	3 (7.3%)	1 (6.7%)	2 (7.7%)	
Indication for tracheostomy or etiology of chronic respiratory failure				
Pulmonary disease	27 (65.9%)	12 (80.0%)	15 (57.7%)	0.488
Central nervous system problem	6 (14.6%)	1 (6.7%)	5 (19.2%)	
Upper airway problem	8 (19.5%)	2 (13.3%)	6 (23.1%)	
Duration from respiratory failure to tracheostomy (days)	26.4 ± 19.9	29.8 ± 24.1	24.4 ± 17.3	0.547
SOFA score	4.6 ± 2.7	5.5 ± 3.3	4.0 ± 2.1	0.211
Laboratory Data				
White blood cell (1000/uL)	9.3 ± 5.8	8.1 ± 3.4	10.0 ± 6.7	0.659
Platelet (1000/uL)	217.2 ± 116.2	181.3 ± 130.0	237.9 ± 104.5	0.086
PT/INR	1.2 ± 0.1	1.2 ± 0.1	1.2 ± 0.1	0.565
Total bilirubin (mg/dL)	0.7 ± 0.9	0.7 ± 0.6	0.7 ± 1.0	0.547
Creatinine (mg/dL)	1.2 ± 1.2	1.1 ± 1.1	1.3 ± 1.3	0.529
Blood gas data before tracheostomy				
P_a_O_2_/F_I_O_2_ ratio (mmHg)	313.5 ± 100.4	294.1 ± 105.2	324.8 ± 97.8	0.277
P_a_CO_2_ (mmHg)	45.3 ± 13.1	48.0 ± 15.3	43.7 ± 11.7	0.445
pH	7.4 ± 0.1	7.4 ± 0.1	7.4 ± 0.1	0.183
Ventilator settings before tracheostomy				
Positive end-expiratory pressure (cm H_2_O)	8.0 ± 0.9	8.4 ± 0.8	7.8 ± 0.9	0.131
Peak airway pressure (cm H_2_O)	22.7 ± 7.0	22.5 ± 5.8	22.8 ± 7.8	>0.999
Tidal volume/predicted body weight (mL/kg)	8.4 ± 2.0	8.7 ± 2.0	8.1 ± 1.9	0.283
Procedure duration				
Bronchoscopy (second)	49.8 ± 43.8	77.3 ± 57.3	33.1 ± 20.6	0.001 *
Tracheostomy (second)	102.8 ± 34.6	98.9 ± 39.0	105.5 ± 31.9	0.334

ICU: intensive care unit; SOFA score: sequential organ failure assessment score; PT/INR: prothrombin time/international normalized ratio. *: *p* Value < 0.05.

**Table 2 diagnostics-13-01764-t002:** Comparison of blood gas data and ventilator setting before and after tracheostomy.

	Before Tracheostomy	After Tracheostomy	*p* Value
PaO_2_/FiO_2_ ratio (mmHg)	313.5 ± 100.4	331.6 ± 93.8	0.196
PaCO_2_ (mmHg)	45.3 ± 13.1	45.4 ± 12.0	0.933
pH	7.4 ± 0.1	7.4 ± 0.1	0.927
Positive end-expiratory pressure (cm H_2_O)	8.1 ± 0.8	7.9 ± 1.0	0.183
Peak airway pressure (cm H_2_O)	23.4 ± 6.9	22.6 ± 7.8	0.403
Tidal volume/predicted body weight (mL/kg)	8.3 ± 1.9	8.3 ± 2.7	0.962

PaO_2_: Partial pressure of oxygen; FiO_2_: Fraction of inspired oxygen; PaCO_2_: Partial pressure of carbon dioxide.

**Table 3 diagnostics-13-01764-t003:** Clinical outcomes in patients with no intra-airway abnormality or different intra-airway abnormalities.

	Total	Liberated from Ventilator	Ventilator Dependent	In-Hospital Mortality
No abnormality	26	13 (50.0%)	9 (34.6%)	4 (15.4%)
Increased secretion	7	4 (57.1%)	1 (14.3%)	2 (28.6%)
Granulation tissue without airway obstruction	3	2 (66.7%)	0 (0.0%)	1 (33.3%)
Mucosal patches	1	0 (0.0%)	1 (100.0%)	0 (0.0%)
Mass lesions with airway obstruction	2	0 (0.0%)	2 (100.0%)	0 (0.0%)
Tumor invasion	1	0 (0.0%)	0 (0.0%)	1 (100.0%)
Peripheral airway mucosal blood oozing	1	0 (0.0%)	1 (100.0%)	0 (0.0%)

## Data Availability

The datasets used and/or analyzed during the current study are available from the corresponding authors upon reasonable request.

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
