# Peer review of "Bronchoscopy Findings during Percutaneous Dilation Tracheostomy: A Single Tertiary Medical Center Experience"

_diagnostics, 2023, doi:10.3390/diagnostics13101764_

Round 1

Reviewer 1 Report

This article investigated bronchoscopy findings during percutaneous dilation tracheostomy in 41 patients. The study was appropriately conducted. I have no major concerns, only a few comments to improve the article. 

The term ‘gender’ should be replaced with ‘sex’ to reflect the biological entity of the patients.

Duration of both tracheostomy and bronchoscopy had quite wide standard deviations. It would be appropriate to add the minimum and maximum amount of time taken of these two procedures and report them as ranges, e.g. 10-100 seconds.

The word “Patients” in the two bottom most boxes in Figure 1 appear larger than the surrounding text. Please edit.

The word “Patients” in line 193 should not have a capitalized “P”.

When discussing tracheostomy, it is worth mentioning the presence of thyroid ima artery, a variant artery supplying the thyroid which exists in 3.8% of the population (https://doi.org/10.1016/j.aanat.2021.151803).

Quality of the writing is satisfactory. 

Reviewer 2 Report

Dear authors,

I have carefully read your study "Bronchoscopy Findings During Percutaneous Dilation Tracheostomy." I thank the editors for inviting me to participate in this review.

The paper is appropriately structured, and the data analyzed are consistent with the aim of the study. However, it is challenging to attribute particular interest to the clinical implications of your scientific communication.

The results you reported in the paper do not overturn the current evidence regarding PDTs and adjuvant intra-operative bronchoscopies. On the other hand, you have no data that would unquestionably strengthen any of the studies already in the literature. In conclusion, you need to get the substantive message across.

My advice is to change the idea of the message by stressing some fundamental points that are often underestimated:

1) should bronchoscopy be done routinely before any PDT procedure, or should it be reserved only for suspected cases after CT or other clinical evidence?

2) how important is the bronchoscopist's skill because of the apparent relationship between bronchoscopy duration and patient gas exchanges?

3) can it be considered always mandated to use operative bronchoscopes during PDT care?

4) what type(s) of PDT device has been used in your center?

5) it would be interesting to verify the clinical outcome of perioperative bronchoscopies in patients undergoing surgical tracheostomy.

6) consider changing the title to indicate to the reader that the study refers to a single-center experience.

Best regards.

Typos and language constructions must be corrected to make the paper more pleasant and smooth for the readers.

Round 2

Reviewer 2 Report

No issues were detected.